# Performance Enhancement of Proton Exchange Membrane Fuel Cell through Carbon Nanofibers Grown In Situ on Carbon Paper

**DOI:** 10.3390/molecules28062810

**Published:** 2023-03-20

**Authors:** Chang Liu, Shang Li

**Affiliations:** 1State Key Laboratory of Advanced Technology for Materials Synthesis and Processing, Wuhan University of Technology, Wuhan 430070, China; 2Foshan Xianhu Laboratory of the Advanced Energy Science and Technology Guangdong Laboratory, Foshan 528200, China

**Keywords:** proton exchange membrane fuel cell, gas diffusion layer, in situ growth carbon nanofibers, microporous layer, surface modification

## Abstract

We developed an integrated gas diffusion layer (GDL) for proton exchange membrane (PEM) fuel cells by growing carbon nanofibers (CNFs) in situ on carbon paper via the electro-polymerization of polyaniline (PANI) on carbon paper followed by a subsequent carbonization treatment process. The CNF/carbon paper showed a microporous structure and a significantly increased pore volume compared to commercial carbon paper. By utilizing this CNF/carbon paper in a PEM fuel cell, it was found that the cell with CNF/carbon paper had superior performance compared to the commercial GDL at both high and low humidity conditions, and its power density was as high as 1.21 W cm^−2^ at 100% relative humidity, which is 26% higher than that of a conventional gas diffusion layer (0.9 W cm^−2^). The significant performance enhancement was attributed to a higher pore volume and porosity of the CNF/carbon paper, which improved gas diffusion in the GDL. In addition, the superior performance of the cell with CNF/carbon paper at low relative humidity demonstrated that it had better water retention than the commercial GDL. This study provides a novel and facile method for the surface modification of GDLs to improve the performance of PEM fuel cells. The CNF/carbon paper with a microporous structure has suitable hydrophobicity and lower through-plane resistance, which makes it promising as an advanced substrate for GDLs in fuel cell applications.

## 1. Introduction

The increase in energy consumption worldwide has raised public awareness about the need for environmental protection and the existing nature of fossil fuels, leading to much research focused on renewable energy sources [1]. However, renewable energy sources, such as solar and wind energy, are unstable and intermittent, making it challenging to use them steadily and constantly; hence, the additional employment of energy storage/generators is needed to improve the utilization rate and stability of renewable energy [2]. Compared to other electrical energy storage technologies, fuel cell power generation is much simpler. Fuel cells generate electricity cleanly and effectively by utilizing the chemical energy of hydrogen or other fuels. They can operate at higher efficiencies than combustion engines and can convert the chemical energy in the fuel directly to electrical energy with efficiencies exceeding 60% [3]. Proton exchange membrane fuel cells (PEMFCs) have received global attention in recent years as alternative power sources for portable, stationary, and automotive applications due to their high power densities, low pollution, and low noise [4,5,6]. The membrane electrode assembly (MEA) is a key part of PEMFCs and consists of a polymer electrolyte membrane, catalyst layers, and gas diffusion layers (GDLs). As a crucial component in PEMFCs, GDLs play primary roles in providing electron conduction, transporting reactants and products, transporting heat, and providing mechanical support for the catalysts [7,8]. Therefore, the ideal GDL has high electrical/electronic conductivity, good gas and water transport characteristics, high thermal conductivity, and high mechanical stability [9].

The GDL consists of a macroporous substrate (either carbon-based or metal-based) and a thin microporous layer (MPL) comprising carbon powder and hydrophobic/hydrophilic agents to provide a flat contact surface with the catalyst layer [10]. Carbon-based materials (carbon paper or carbon cloth) are the most commonly used macroporous substrates due to their high stability in acid environments, high gas permeability, good electronic conductivity, and elastic compression properties [10,11]. The mass transport limitation caused by liquid water, especially at high current densities, is one of the key factors affecting the performance of PEMFCs. The difficulty in removing water from the cathodes of PEMFCs leads to compromised oxygen transfer to the reaction sites at the cathode electrodes [12,13]. Water flooding is caused by a water removal rate that is lower than the generation rate at cathodes, which hinders oxygen transport by blocking the pores between the catalyst layer and GDL. Water flooding within the catalyst layer, GDL, and flow field can result in nonuniformity of the distribution of reactants in the catalyst layer, thus leading to the poor performance of PEMFCs [12].

In order to effectively remove liquid water and avoid flooding at the cathode of PEM fuel cells [14], several surface modification methods, including adding hydrophobic agents and laser perforation [15,16] on carbon-based GDLs, have been utilized to improve the water management and cell performance. For example, adding hydrophobic agents, such as polytetrafluoroethylene (PTFE) [17,18,19], fluorinated ethylene propylene (FEP) [20], or perfluoropolyether (PFPE) [21] on carbon paper or cloth can increase the hydrophobicity of the GDL. However, the addition of a hydrophobic agent in the GDL may reduce water saturation, causing poor gas transport and high electronic resistance. Liu et al. [10] reported on a novel hydrophilically modified GDL developed by inserting polyacrylonitrile (PAN) into the microporous layer. The power density of the PEM fuel cell with PAN on the GDL was 30% higher (0.616 W cm^−2^) than that of a conventional hydrophobic GDL (0.480 W cm^−2^) at low humidity. They attributed the improvement in the water retention ability of the PAN and, thereby, the increase in the degree of membrane hydration to the GDLs. Another study by Lim et al. [22] reported a GDL modified via the deposition of HfO_2_ on the MPL, which had a higher power density than the reference commercial GDL due to better water management and lower charge transfer resistance and mass transport resistance. Though many efforts to modify the surface of the GDL have been made, most focused on the MPL rather than directly on the carbon paper.

Carbon nanofibers (CNFs) and carbon nanotubes (CNTs) can be used as catalyst supports [23,24,25,26,27] and grown in situ on carbon paper as GDLs or microporous layers (MPLs) in PEM fuel cells. Due to their electrical conductivity, thermal conductivity, mechanical strength, and increased surface area, they are widely used in PEM fuel cell applications [28,29,30,31]. A few studies have demonstrated that the in situ-grown CNT/CNF layer on carbon paper is beneficial for improving the PEMFC’s performance. Gao et al. reported on the development of a carbon nanotube-based gas diffusion layer (CNT-GDL) via sintering wet carbon paper made of CNTs, polyacrylonitrile-based carbon fiber (PCF), and PTFE for direct methanol fuel cells (DMFCs). The cell with a CNT-GDL showed a 27% improvement in peak power density due to its enhanced electrical conductivity and mass transfer ability compared to a Toray GDL [32]. In addition, it has been reported that several different precursor materials can be used for this purpose, such as PCF, rayon-based graphite felt (RGF), and pitch-based graphite felt (PGF) [33]. When the thickness and porosity are the same, PCF has a higher conductivity and lower ohmic polarization loss, while rayon-based PCF has a higher surface roughness [34]. Celebi et al. developed a carbon paper grown with carbon nanofibers (CNFs) via the homogeneous deposition precipitation of nickel hydroxide, which improved its water management as a gas diffusion medium [29]. Maheshwari et al. [35] modified a carbon paper by coating the non-functionalized carbon fiber with multi-walled carbon nanotubes (MWCNTs). The modified carbon paper performed better than the non-modified carbon paper. Xie et al. [30] reported that CNTs grown in situ on carbon paper as a gas diffusion layer reduced water flooding and promoted the mass transfer of the PEMFC by ensuring suitable hydrophobicity and a proper structure.

Polyaniline (PANI) is one of the most interesting electro-conducting polymers due to its diverse applications, such as its use as a surface-modified electrode and its ability to promote chemical and biological sensing (due to its excellent characteristics and the low price of monomers) [36,37,38,39]. CNTs and CNFs modified with PANI have been proposed for applications in supercapacitors [40,41] and sensors [42]. It has been reported that PANI-modified CNFs and CNTs can serve as a dispersant and stabilizer to immobilize Pt nanoparticles, leading to enhanced surface properties, such as improved electrical conductivity, chemical resistance, and increased surface area in PEM fuel cells [43,44]. Moreover, it has also been reported that PANI-decorated carbon-material-supported Pt nanoparticles improve the electrode stability in fuel cells [45,46]. The conversion of conducting polymers into carbonaceous structures is a promising method to link both types of materials. As a nitrogen-containing polymer, PANI is suitable for carbonization and thermal treatment to obtain carbonaceous materials, i.e., CNTs and CNFs [47,48].

In this study, CNF grown in situ on carbon paper as an integrated GDL was developed using electro-polymerization of PANI directly on the carbon paper and a subsequent carbonization treatment process to improve the performance of a PEM fuel cell. The properties of CNF/carbon paper, i.e., its surface morphology, porosity, pore size distribution, contact resistance, and water contact angle, were studied. The performance of the PEM fuel cell with the CNF/carbon paper was compared to that of a commercial GDL. The single-cell test results reveal that the cell with the CNF/carbon paper had an improved performance under both high- and low-humidity conditions. This study provides a novel surface modification method for carbon paper, which could be promising as an advanced substrate for GDLs in fuel cell applications.

## 2. Results and Discussions

### 2.1. Surface Morphology

Figure 1 shows the SEM images of the pristine carbon paper, hydrophobic commercial carbon paper, and the CNF/carbon paper prepared in this work at 3000× and 30,000× magnification. Figure 1b shows that some pores inside the hydrophobic commercial carbon paper were blocked by a substance, which is supposed to be the hydrophobic agent. Compared with Figure 1a,b, the CNFs on carbon paper showed a mesh and microporous structure, and the diameters of the CNFs ranged from approximately 50 to 100 nm. The presence of the CNFs on the carbon paper changed the surface morphology of the carbon paper and had further effects on the hydrophobicity of the carbon paper.

### 2.2. Porosity and Pore Size Distribution

Table 1 provides data on the porosity of the hydrophobic commercial carbon paper and the CNF/carbon paper. The porosity of the CNF/carbon paper was much higher (82.8%) than that of the hydrophobic commercial carbon paper (76.5%), which corresponds well to the results shown in Figure 1. For commercial carbon paper, some pores were blocked due to the addition of the hydrophobic agent, reducing the porosity of the carbon paper. However, the CNF/carbon paper maintained its high porosity after surface modification via electrochemical deposition and subsequent carbonization. In general, high porosity indicates high gas permeability, which promotes gas diffusion in the GDL.

The pore size distribution is shown in Figure 2. It can be seen that, for both carbon papers, the vast majority of pores had diameters of 20 to 60 μm. Macro pores can prevent the condensation of liquid water, which effectively inhibits flooding of the MEA and enhances mass transport in the GDL. However, when compared to commercial carbon paper, the CNF/carbon paper exhibited a higher ratio of the volume of the macro pores to the entire pores, which demonstrates that some pores in the commercial carbon paper were blocked, making its mass transport ability weaker than that of CNF/carbon paper.

### 2.3. Contact Resistance

In this work, the through-plane resistance of the carbon papers was conducted. Figure 3a displays the through-plane resistance of the samples using a conventional press setup. The carbon paper was placed between two highly conductive plates (gold-coated copper plates) under a defined compression. The sample was then exposed to a four-wire resistance measurement with the DC. The plate-to-plate voltage drop was measured to calculate the resistance of the carbon papers [49,50,51]. The through-plane resistance includes the contributions from the bulk material and the two contact resistances between the carbon paper and plates.

The through-plane resistances of the hydrophobic commercial carbon paper and CNF/carbon paper are shown in Figure 3. The through-plane resistance for both carbon papers decreased with the increase in pressure. The carbon papers became denser with the increase in the pressure, increasing the number of contact points of the carbon fibers inside the carbon papers. Meanwhile, the carbon papers became thinner, which shortened the distance for electron transport and caused a decrease in the through-plane resistance with increasing pressure. The contact resistance of the CNF/carbon paper was lower than that of the hydrophobic commercial carbon paper over the entire pressure range. This might have been caused by the presence of the nonconductive hydrophobic agent, which bound the carbon fiber and reduced the length of the pathway for the electron transport in the commercial carbon paper.

### 2.4. Water Contact Angle

The water contact angle of GDLs has a significant impact on water management, which can be used to estimate the hydrophobicity of a GDL. Water can be promptly expelled from GDLs with appropriate hydrophobicity without blocking the gas diffusion channel [30]. Figure 4 shows the contact angle for water droplets with different carbon papers. The commercial carbon paper had the highest contact angle of 150° due to the presence of a hydrophobic agent, which significantly increased the hydrophobicity. The contact angle of the pristine carbon paper was 126°. The CNF/carbon paper had a higher contact angle (138°) due to the intrinsic hydrophobic property of the CNF, indicating that the hydrophobicity of the GDL was modified via growing CNF in situ on it. In addition, the contact angle was not only affected by the composition of CNF but also by its specific surface area and roughness. The higher contact angle of the CNF/carbon paper might be mainly attributed to a coarser surface roughness and higher specific surface area, which can be seen in Figure 1c,d. Additionally, it can be deduced from comparing the contact angles that the CNF/carbon paper had better water retention than the commercial carbon paper. It has been reported that PEM fuel cells with a hydrophilic additive on the GDL do not show much difference in performance at high relative humidity (RH) (e.g., 100% RH); however, they do improve the cell performance when the RH is low [10]. Here, the commercial carbon paper showed a higher contact angle than the CNF/carbon paper, which means it is more hydrophobic. Though high hydrophobicity is advantageous for removing water from the GDL, a hydrophilic additive, such as CNF, improves the water retention ability of the GDL, increasing the degree of membrane hydration, especially at a low RH. 

### 2.5. Fuel Cell Testing

Figure 5 displays the results of the single-cell test for commercial carbon paper and CNF/carbon paper. The performances of these two cells were similar when the current density was lower than 1000 mA cm^−2^; however, when the current density increased beyond 1000 mA cm^−2^, the performance of the cell assembled with CNF/carbon paper was superior to that of the cell assembled with commercial carbon paper. Figure 5a shows that the power density of the CNF/carbon paper was 1.21 W cm^−2^ at 100% RH, which is ~26% higher than that of the commercial GDL at 0.9 W cm^−2^. This is because the CNF/carbon paper had a higher porosity, which is beneficial for the diffusion and transport of the gas. A recent study by Liu and Shao et al. [10] reported the development of a novel hydrophilically modified GDL by inserting polyacrylonitrile (PAN) into the microporous layer. The cell with PAN on the GDL exhibited a maximum power density of 0.616 W cm^−2^, which is 30% higher than that of the conventional hydrophobic GDL (0.480 W cm^−2^) under low humidity. They attributed this significant enhancement in the performance to the powerful wettability and the pore structure modification of the PAN. Our study shows that the power density of the cell with CNF/carbon paper was 1.12 W cm^−2^ at 40% RH, which is significantly higher than that of the cell with the commercial GDL (0.7 W cm^−2^) and the modified GDL reported in their study. Additionally, the difference in cell voltage between the cell with the commercial GDL and CNF/carbon paper increased with decreasing relative humidity under high-current-density conditions. For example, the difference in the voltage was 0.142 V at 100% RH, 0.149 V at 70% RH, and 0.202 V at 40% RH at 2100 mA cm^−2^. The superior performance of the CNF/carbon paper at low RH was due to its water retention ability, which was greater than that of the commercial carbon paper, as can be observed in the contact angle results (Figure 4). At low relative humidity, the CNF/carbon paper retained more water in the GDL, which made the real relative humidity in the MEA higher than the set value and finally improved the cell performance at low relative humidity conditions. In addition, the pore structure modification of CNF made the surface of the GDL flatter, resulting in better water management of the fuel cell. The lower through-plane resistance of the CNF/carbon paper may have also contributed to its improved cell performance, which is in accordance with the results presented in Figure 3.

## 3. Materials and Methods

### 3.1. Fabrication of CNF/Carbon Paper

Electro-polymerization was performed on a CHI 660 A workstation using carbon paper (Toray Industry) (area = 25 cm^2^). The carbon paper was clamped using a Pt electrode holder as the working electrode (WE), a platinum black electrode as the counter electrode, and a saturated calomel electrode (SCE) as the reference electrode (RE). Electrochemical polymerization was performed over 10 cycles, scanning back and forth from 1 to 3 V vs. SCE at a scan rate of 50 mV/s in 0.6 M HClO_4_ aqueous solution (Sinopharm Chemical Reagent Co., Ltd., Shanghai, China) containing 0.2 M aniline (Aladdin Industrial Corporation, Shanghai, China). The solutions were prepared using deionized water obtained from a lab-deionized water filter with a resistivity of 18.25 MΩ cm^−1^.

After drying in an oven at 60 °C for 4 h, the carbon paper coated with PANI nanofibers was then placed in a tube furnace under N_2_ (99.999% purity) with a flow rate of 150 mL min^−1^ for 1 h to remove any residual air. Then, the tube furnace was adjusted to increase the temperature at a ramping rate of 5 °C min^−1^ until the heat treatment temperature reached 1000 °C, and the temperature was then maintained for 2 h. The obtained CNF/carbon paper was coated with MPL (Wuhan WUT New Energy Co., Ltd., Wuhan, China) on one side of the carbon paper. In this way, the final CNF/carbon paper was obtained. A schematic of the preparation of CNF/carbon paper is shown in Figure 6.

### 3.2. Characterization

Scanning electron microscopy (SEM, Zeiss Ultra Plus, Zeiss, Jena, Germany) was used to evaluate the micro-structural surface morphologies of the CNF/carbon paper and commercial carbon paper. The porosity and pore size distribution were measured using the mercury intrusion porosimeter (MIP) method (AutoPore IV 9500, Micromeritics Instrument Corp., Atlanta, GA, USA). The hydrophobicity of the CNF/carbon paper was evaluated by measuring the contact angle of the water (Theta Lite, Biolin Scientific, Västra Frölunda, Sweden).

The GDL was compressed between two conductive, gold-coated plates to measure through-plane resistance. The sample was subsequently subjected to a four-wire resistance test using the DC. The through-plane resistance of the GDLs, which is the sum of the GDL resistance and the two contact resistances that are present between the tested GDLs and gold-coated plates, was then determined by measuring the plate-to-plate voltage drop [49,50]. The primary factor affecting conductivity in the through-plane direction was contact resistance [52].

### 3.3. Single-Cell Components and Performance Measurements

The commercial catalyst coated membrane (CCM) (15 μm thick Nafion membrane from WL Gore & Associates, Inc., Flagstaff, AZ, USA, 0.1/0.4 mg cm^−2^ Pt/C for the anode and cathode, respectively) was assembled with the obtained CNF/carbon paper at the cathode and the commercial GDL (JNT20-A3, JNTG with MPL, 250 µm thick, 78% porosity, and 10% PTFE treatment) at the anode. As a reference, the same type of CCM was sandwiched by commercial GDLs at both the cathode and anode for comparison. One side of the obtained CNF/carbon paper was coated with MPL (Wuhan WUT New Energy Co., Ltd.). The coating method and MPL composition used for the commercial GDL were the same as those used for CNF/carbon paper.

The single-cell measurements were taken in a PEM fuel cell test station (HEPHAS energy, HTS-125, Hsinchu, Taiwan). The geometrically active area of the single-cell test was 25 cm^2^, and the test was carried out in humidified H_2_/air gas at a flow of 200 mL·min^−1^ and 500 mL·min^−1^ for the cathode and anode, respectively. The test was conducted at 80 °C under atmospheric pressure. The relative humidity used for the anode was 100%, while that used for the cathode varied between 40%, 70%, and 100%. The polarization curves were measured under a constant current using a 100 mA cm^−2^ current step and a 1 min dwell time. The stoichiometries of hydrogen and air were 1.5 and 2.5, respectively.

## 4. Conclusions

In this study, carbon nanofibers grown in situ on carbon paper to create an integrated GDL for PEM fuel cells were developed via the electro-polymerization of PANI on carbon paper, followed by a subsequent carbonization treatment process. The CNF/carbon paper showed a better cell performance than commercial carbon paper at both low and high humidity conditions due to its high pore volume, high electronic conductivity, and sufficient hydrophobicity, which aided water management inside the GDL. The power density of the cell with the CNF/carbon paper reached 1.21 W cm^−2^ at 100% relative humidity, which is 26% higher than that of the commercial GDL of 0.9 W cm^−2^. The apparent improvement in performance was attributed to the following reasons. On the one hand, the CNF had a higher pore volume and porosity, which improved gas diffusion in the GDL. On the other hand, the CNF made the surface of the carbon paper flatter, resulting in better water management for the fuel cell. Additionally, the superior performance of the cell with CNF/carbon paper at low relative humidity demonstrates that it had better water retention than the commercial GDL. The GDLs with suitable hydrophobicity and a proper pore structure could reduce water flooding at the cathode side, especially at high current density. Our findings show that the novel CNF/carbon paper developed in this study is promising as an advanced substrate of the GDL for fuel cell applications in both low and high humidity conditions. Future studies will be carried out to optimize the properties and structures of the CNF/carbon paper, e.g., by adjusting the deposition time of electro-polymerization and carbonizing temperature of PANI. The durability of the PEM fuel cell with a CNF/carbon paper will be investigated in the future.

## Figures and Tables

**Figure 1 molecules-28-02810-f001:**
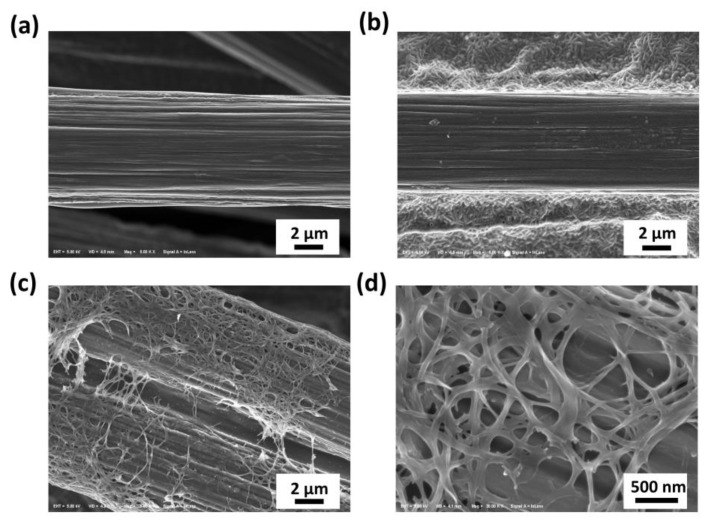
SEM images showing (**a**) pristine carbon paper; (**b**) hydrophobic commercial carbon paper; (**c**,**d**) CNF/carbon paper at 3000× and 30,000× magnification.

**Figure 2 molecules-28-02810-f002:**
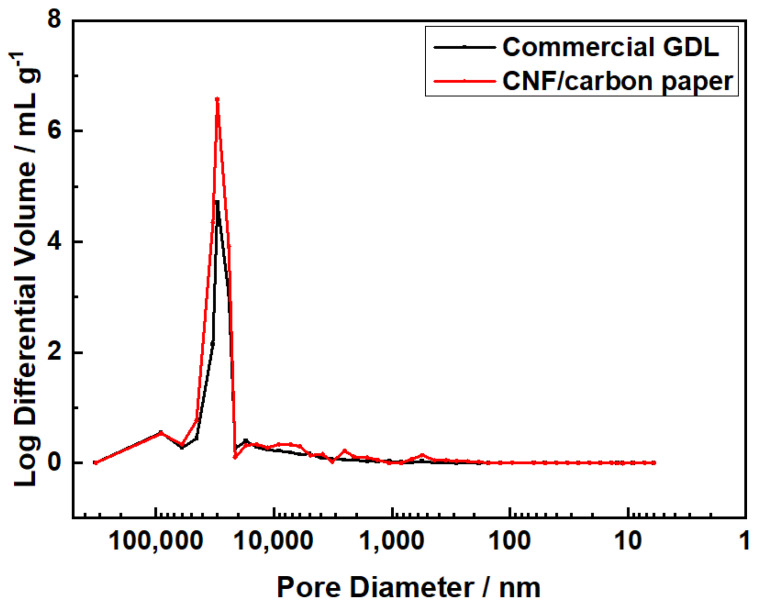
The pore size distribution of hydrophobic commercial carbon paper and CNF/carbon paper.

**Figure 3 molecules-28-02810-f003:**
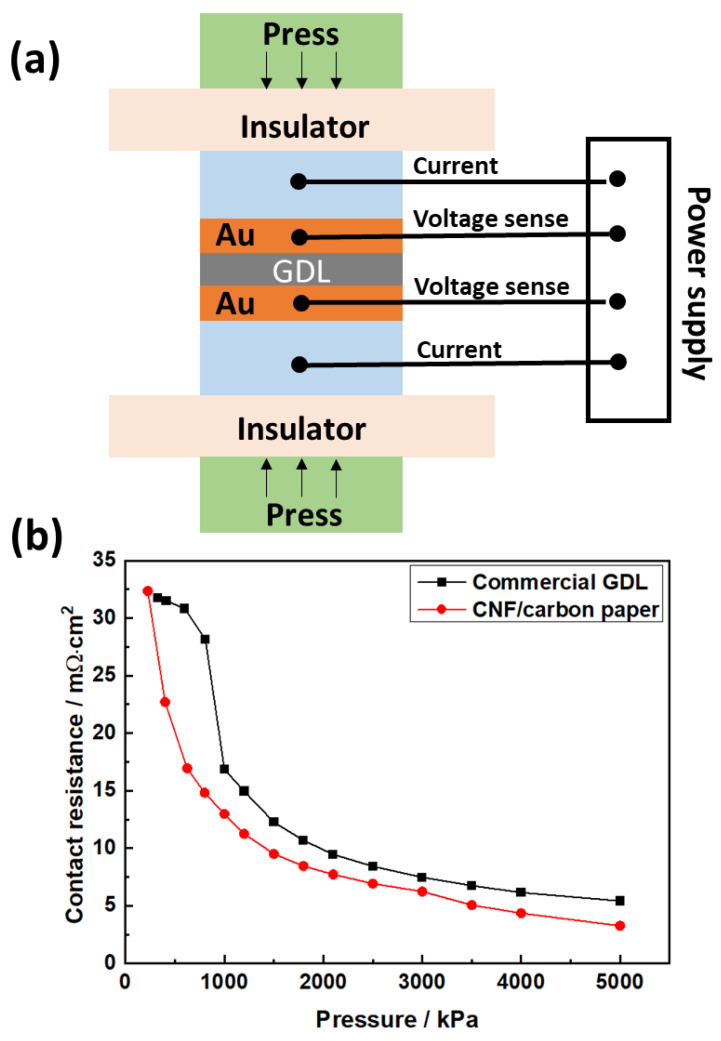
The through-plane resistance of hydrophobic commercial carbon paper (**a**) and CNF/carbon paper as a function of pressure (**b**).

**Figure 4 molecules-28-02810-f004:**
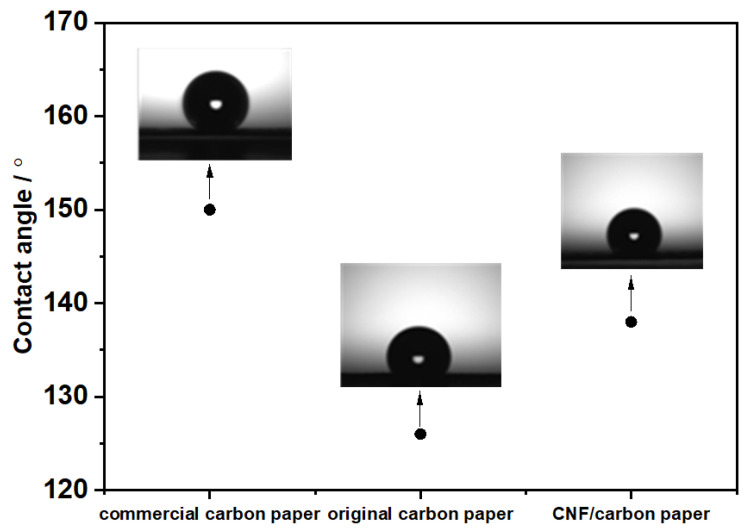
The contact angle of hydrophobic commercial carbon paper, original carbon paper, and CNF/carbon paper.

**Figure 5 molecules-28-02810-f005:**
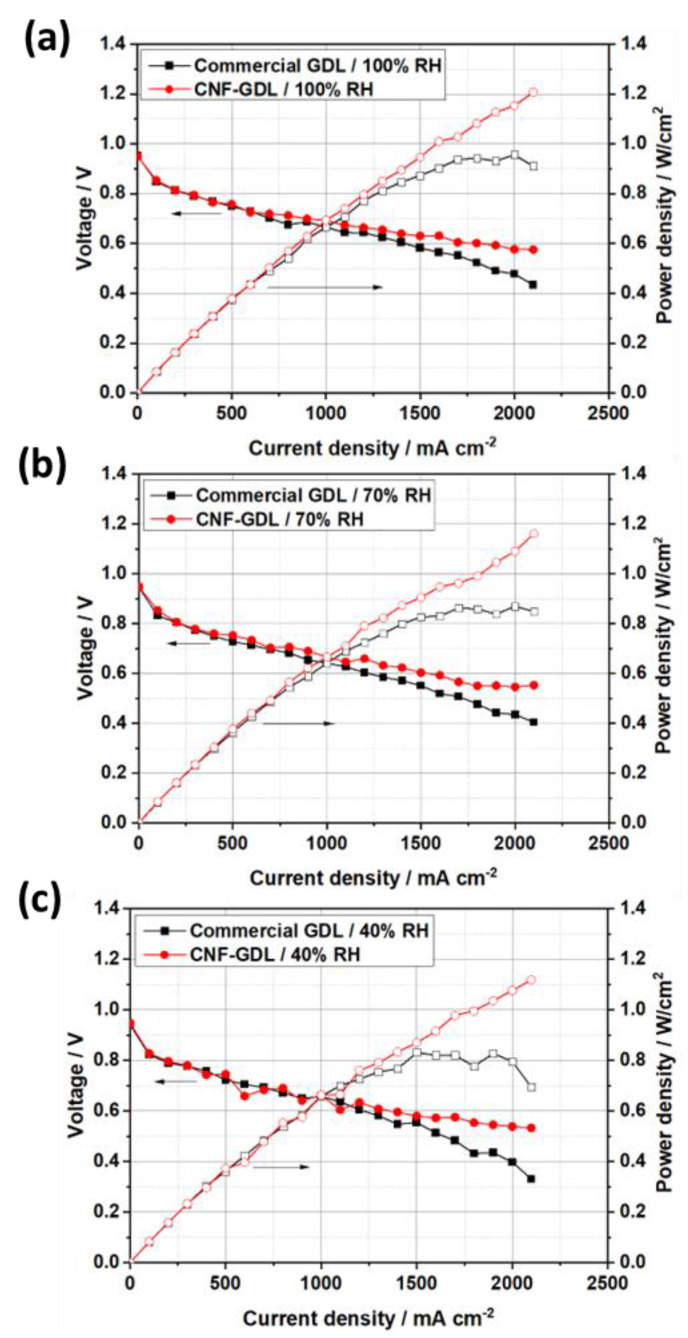
The single-cell performance of hydrophobic commercial carbon paper and CNF/carbon paper at different relative humidity conditions: (**a**) 100% RH, (**b**) 70% RH, (**c**) 40% RH.

**Figure 6 molecules-28-02810-f006:**
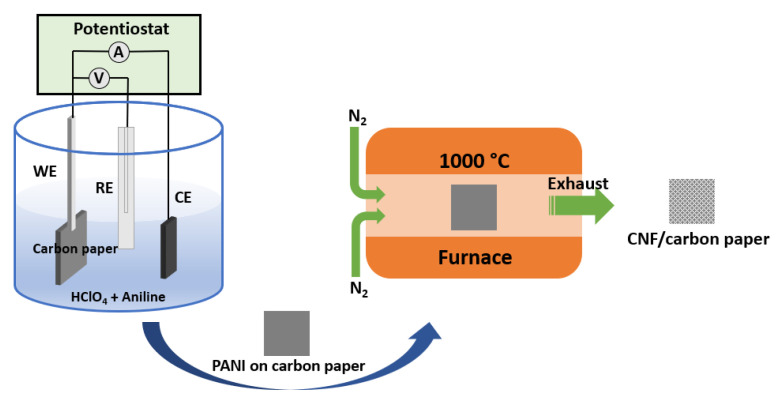
Schematic of the preparation of CNF/carbon paper.

**Table 1 molecules-28-02810-t001:** The porosity of hydrophobic commercial carbon paper and CNF/carbon paper.

Sample	Commercial Carbon Paper	CNF/Carbon Paper
Porosity	76.5%	82.8%

## Data Availability

Not applicable.

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
