# Peer review of "Performance Enhancement of Proton Exchange Membrane Fuel Cell through Carbon Nanofibers Grown In Situ on Carbon Paper"

_molecules, 2023, doi:10.3390/molecules28062810_

Round 1

Reviewer 1 Report

In this work, the author used a new method for GDL surface modification. The modified GDL has a higher porosity and lower electronic resistance compared to commercial GDL. The MEA testing results show that the cell has better performance, especially at high current density regions, and both at low and high relative humidities. In general, this work is very interesting to the fuel cell community. However, several issues need to be resolved before they can be published. 

1. several important experiment details are missing. For example, what is the membrane used for the MEA test? What is the thickness of gaskets used for the fuel cell tests, and the compression ratio for anode and cathode GDL? The author mentioned H2/Air gas flow rate for single-cell testing, but also said the stoichiometric ratio is fixed at 1.5 and 2.5, which is confusing. Is the author using a constant flow rate for all current densities? Or it is actually constant stoic?

2. The author is comparing the CNF-modified carbon paper coated with an extra layer of MPL, with commercial GDL that already contains an MPL layer. Is it possible that the fuel cell performance difference actually comes from the difference in MPL coating quality? Are those two MPL layer coated with the same method, and has the same composition? 

Author Response

Dear reviewer, 

Thanks for the great comments for the paper. Here are the answers about the questions:

  1. The MEAs were ordered from Wuhan WUT New Energy Co. Ltd company. The membrane is 15 μm thick Nafion membrane from WL Gore & Associates, Inc. The stoichiometries of hydrogen and air were  constant at 1.5 and 2.5.
  2. Both the CNF/carbon paper and commercial GDL used the same MPL from Wuhan WUT New Energy Co. Ltd company. The coating method and composition are totally the same.

The details of the materials and experiments were added in the experimental section.

Reviewer 2 Report

This work could be reconsidered after addressing carefully the following comments:

1.The title of the manuscript should be more sharp.

2.The abbreviations should not be used in the title, abstract, and keywords.

3.The motivation and contribution of this works must be clarified in the introduction section.

4.Some quantities results should be presented in abstract and conclusion.

6.The ideas/criteria of choosing the size/arrangement of considered system should be given?

7.More explanation is required in results and discussion section.

8.For more contribution, the authors should compare their results with the related results in other published works.

9.There are many grammatical errors and typoerros throughout the whole manuscript? The paper should be rechecked.

10.The novelty of the work must be clearly addressed and discussed, compare your research with existing research findings and highlight novelty, (compare your work with existing research findings and highlight novelty).

11.Conclusion: Future scope of the work should be provided.

12.The literature review section is very weak. As they are many recent published papers on the same topic need to be included.

Author Response

Dear reviewer,

Thanks,

Chang

Reviewer 3 Report

This paper proposes the in-situ growth of CNFs on carbon paper as integrated GDL for PEMFC by the electro-polymerization of polyaniline and subsequent carbonization treatment process. The CNF/carbon paper shows a remarkably larger pore volume compared to commercial carbon paper. By utilizing this CNF/carbon paper in PEMFC, it is found that the cell with CNF/carbon paper showed higher performance than commercial GDL, and it performed much better in a relatively low humidity environment. This study provides a novel and facile method for surface modification of GDL to improve the performance of PEMFCs. 

The content of the manuscript fits the research scope of the journal, and will also arouse the wide reading interest of the journal readers. However, the manuscript needs some improvements. The results and discussion section still need to be improved as well. Therefore, I decided to give a minor revision, hoping that the author can make targeted improvements and promotions according to the suggestions put forward by the reviewer.

1. The quantity of the keywords should be enough to attract readers' interest in reading. Currently, there are few keywords and there is no good summary of the content of the manuscript. Therefore, I suggest adding keywords appropriately, such as 'in-situ growth', 'carbon paper', 'high power density', and 'microporous'.

2. Page 1, 'Proton exchange membrane fuel cells (PEMFCs) attract great interest as alternative power sources for portable, stationary and automotive applications due to their high power densities, low pollution and low noise [1–3].'

    This sentence is a bit abrupt, especially about what is a fuel cell and why should fuel cells be used? It is suggested to supplement relevant content appropriately. For example, by using PEFC, we can directly convert the chemical energy of fuels to electrical energy, so the thermal loss by the combustion process can be avoided. In the context of global decarbonization, this technology has received increasing attention. 

    Moreover, renewable energies are also very popular under the global decarbonization background. What are the advantages of fuel cells over renewable energy? This should also be briefly explained as well. For example, renewable energy sources such as solar/wind energy are unstable and intermittent during generation, and the generated electric energies are difficult to apply continuously and stably. Hence, the additional employment of energy storage/generators is needed to improve the utilization rate and stability of renewable energy (ChemSusChem, 2022, 15(1): e202101798). The operation of fuel cell power generation is obviously simpler.

3. Page 2, 'Gao et al. reported a carbon nanotube-based gas diffusion layer (CNT-GDL) by sintering the wet carbon paper made of CNT, polyacrylonitrile-based carbon fiber (PCF) and PTFE for direct methanol fuel cells (DMFC).' Here refers to polyacrylonitrile-based PCF, there is also rayon-based PCF. What is the difference between the polyacrylonitrile and rayon-based PCF and how about their influences on the performance?  It should be explained a bit better. At the same thickness and porosity, PAN-based PCF has higher conductivity and smaller ohmic polarization loss, while rayon-based PCF has a higher surface roughness (10.1002/celc.201900518).

4. Page 2, 'Polyaniline (PANI) is one of the most interesting electro-conducting polymers for different applications such as surface-modified electrodes and chemical and biological sensing due to the excellent characteristics and low price of the monomer [30–33].' How about the stability of PANI? Is it water-soluble? Is it stable in an acidic environment, since protons are generated during the PEMFC operation? These issues should be clarified. 

5. Page 6, 'High hydrophobicity is advantageous for removing water from the GDL, but it may cause water to accumulate at the GDL/catalyst layer interface, increasing the transfer re-sistance for oxygen transportation.' This sentence is inconsistent. If it is good for drainage from the GDL, how can the oxygen be blocked and water accumulate again at the GDL/catalyst layer interface? The catalyst should also be hydrophobic, and nothing there is hydrophilic. This issue should be better explained to convince the readers.

6. The experimental design part of the whole manuscript is too simple, such as the preparation part, the authors should get a series of samples, CNF/CP, instead of one sample. In the process of preparing CNF/CP composite electrodes, it is easy to obtain a series of samples by appropriately changing the preparation process and parameters. Comparing these samples and optimizing the components of CNF/CP will make this work more meaningful.

7. The characterization is also rather simple, for example, how about the thickness and perforation diameter influence of the GDL layer (10.1016/j.ijheatmasstransfer.2019.05.008)?  In the gas diffusion layer, the transfer of reaction gas, the transfer of reaction products and the transfer of electrons are mainly carried out. Investigating the properties of the diffusion layer is to mainly examine the transmission capabilities of these three aspects. Generally, in addition to directly analyzing the properties of the diffusion layer from the polarization curve, some physical means have also been established to characterize the properties of the diffusion layer, mainly including the fluid transport characteristics, electrical conductivity, pore structure, and hydrophilic/hydrophobic properties of the diffusion layer. These parts should be further expanded, and now it seems not comprehensive enough.

Reviewer 4 Report

I would not recommend this article “Carbon Nanofibers In situ Grown on Carbon Paper as Integrated Gas Diffusion Layer for PEM Fuel Cell” in the MDPI journal because of following serious concerns about the article which should be reconsidered.

1.     There are many grammatical and narration mistakes which should be revised. As in the abstract line No. 11 “fuel cell was developed” should be “fuel cells were developed”.

2.     Line no. 13, “comparing” should be “compared”.

3.     Line no. 14, “paper into PEM” should be “paper in PEM” while “it is found” should be “it was found”.

4.     Line no. 16, “better at” should be “better in”.

5.     Line no. 19, “paper with” should be “paper has” while “hydrophobicity property” should be “hydrophobicity properties”.

6.     Line no. 20, “promising to be” should be “promising as” while “substrate of” should be “substrate for”.

7.     Line no. 21, “application” should be “applications”.

8.     Line no. 39, “due to high” should be “due to their high” while “environment” shold be “environments”.

9.     Line no. 43, “transferal of oxygen” should be “oxygen transfer”.

10.  Line no. 50, “agent” should be “agents”.

11.  Line no. 55, “additive of” should be “of”.

12.  Line no. 56, “causes” should be “cause”

13.  Line no. 57, “catalyst support” should be “catalyst supports”.

14.  Line no. 60-61, should be paraphrased to make it more fluent.

15.  Line no. 64, “improvement of” should be “improvement in”.

16.  Line no, 65, “than those of” should be “than those with”.

17.  Line no. 68, “diffusion media” should be “diffusion medium”.

18.  Line no. 70-73 should be rephrased.

19.  Line no. 85-87, “The performance of the PEM fuel cell with the commercial GDL and CNF/carbon paper were compared.” Should be “The performance of the PEM fuel cell was compared with that of the commercial GDL and CNF/carbon paper”.

20.  Line no. 87-88, replace “method of”, “paper which”, “promising to be an”, “substrate of” and “application” with “method for”, “paper that”, “promising to be an”, “substrate for” and “applications” respectively.

21.  Line no. 92, “work with” should be “work at”.

22.  Line no. 93 there is a spacing problem between “Figure 1 (b)”.

23.  Line no. 94, “which” should be “that”.

24.  Line no. 95, “paper shows” should be “paper show”.

25.  Line no. 98, “and further effect on the hydrophobicity of the carbon paper” should be written as “and had a further effect on its hydrophobicity”.

26.  Line no. 104-106, paraphrase these lines.

27.  Line no. 114, “avoid” should be “avoids”.

28.  Line no. 115, “enhance” should be “enhances”.

29.  Line no. 128, “and a defined” should be “and”.

30.  Line no. 134, “the increasing of pressure” should be “the increasing pressure”.

31.  Line no. 136, “with the increasing of pressure” should be “with the increased pressure”.

32.  Line no. 137, “shorten”, “for electrons transport” and “causing the decrease of” should be “shortens”, “for electron transport” and “causes the decrease in” respectively.

33.  Line no 152, “agent of it” and “highly increase” should be “agent in it” and “highly increases” respectively.

34.  Line no. 159, “the value of the contact” should be “the values of the contact”.

35.  Rewrite the whole heading of “Single cell performance” in an effective and precise manner.

36.  Line no. 229, double round brackets must be corrected.

37.  Paraphrase line no. 233-234.

38.  Line no. 242, “was developed” should be “were developed”.

39.  Line no. 264, “analyses” should be “analysis”.

40.  There is punctuation mistake in the “conflict of interest”.

41.  Rewrite the conclusion and add your findings.

42.  The schematic diagram of the process must be added.

43.  The article is not in the order. “Material and methods” must come after introduction, not after “Result and discussion”.

Round 2

Reviewer 2 Report

This work could be accepted in the present modified version.

Reviewer 4 Report

Authors have incorporated all suggestions so this manuscript can be accepted in its present form.